# Immunotherapy and Cancer: The Multi-Omics Perspective

**DOI:** 10.3390/ijms25063563

**Published:** 2024-03-21

**Authors:** Clelia Donisi, Andrea Pretta, Valeria Pusceddu, Pina Ziranu, Eleonora Lai, Marco Puzzoni, Stefano Mariani, Elena Massa, Clelia Madeddu, Mario Scartozzi

**Affiliations:** Medical Oncology Unit, Department of Medical Sciences and Public Health, “Azienda Ospedaliero Universitaria” of Cagliari, University of Cagliari, 09042 Cagliari, Italy; an.pretta@gmail.com (A.P.); valeria.pusce@gmail.com (V.P.); pi.ziranu@gmail.com (P.Z.); ele.lai87@gmail.com (E.L.); marcopuzzoni@gmail.com (M.P.); mariani.step@gmail.com (S.M.); elenamassa@unica.it (E.M.); clelia_md@yahoo.it (C.M.); marioscartozzi@unica.it (M.S.)

**Keywords:** immunotherapy, AI, multi-omics science, biomarkers, state of the art

## Abstract

Immunotherapies have revolutionized cancer treatment approaches. Because not all patients respond positively to immune therapeutic agents, it represents a challenge for scientists who strive to understand the mechanisms behind such resistance. In-depth exploration of tumor biology, using novel technologies such as omics science, can help decode the role of the tumor immune microenvironment (TIME) in producing a response to the immune blockade strategies. It can also help to identify biomarkers for patient stratification and personalized treatment. This review aims to explore these new models and highlight their possible pivotal role in changing clinical practice.

## 1. Introduction

Immunotherapy (IT) represents a significant achievement in cancer treatment [1]. Tumor immunotherapy works by restarting the tumor immune cycle and restoring the body’s natural anti-tumor immune response [2]. Currently, there are at least four main kinds of immunotherapy strategies, which include immune checkpoint inhibitors (ICIs) such as Programmed cell Death protein-1 (PD-1) and Cytotoxic T-Lymphocyte Antigen 4 (CTLA-4), chimeric antigen receptor T-cell therapy, tumor vaccines, and peripatetic immunotherapy. Although these therapies have been widely successful, enhancing clinical oncology outcomes [2], not all patients have benefited from it [1]. Therefore, it is crucial to screen who will gain from immunotherapy the most [2]. Tumor heterogeneity might be the reason underlying a lower treatment efficacy due to several factors such as genetic, epigenetic, and transcriptional modifications; protein expression variations; and changes in metabolic profiles [3].

Lately, there has been a lot of attention given to post-translation modifications (PTMs) which are small changes made to single amino acids such as glycosylation, acetylation, phosphorylation, palmitoylation, and ubiquitination or deubiquitination. These PTMs have been found to have the ability to alter the function, shape, balance, and interaction of proteins with other molecules. Furthermore, recent studies have shown that the expression levels of PD-1 and Programmed cell Death Ligand 1 (PD-L1) can be regulated by epigenetic, transcriptional, and post-transcriptional systems, which, in turn, impact tumor immunity [4,5].

In this scenario, multi-omics approaches, which incorporate genomics, transcriptomics, proteomics, metabolomics, radiomics, and immunomics, help to shed light on the various biological layers present within tumors and explore protein copiousness, metabolic signature athwart disparate cellular types, mRNA expression levels, and genomic modifications with the aim to decode the molecular landscape of cancer along with the tumor–immune interaction mechanisms, identifying new potential biomarkers and targets for immunotherapy, and facilitate the identification of distinctive molecular signatures linked to immunotherapy responders and non-responders in order to personalize treatments and improve patient outcomes [3].

Moreover, temporal multi-omics can be used to tail the dynamic modification inside cancers or induced by the drugs over the course of time, helping to dissect tumor heterogeneity and its influence on therapies [3].

A thorough review of pan-cancer research that evaluates the multi-omics approach and Artificial Intelligence (AI) methodology, along with the latest data translated into clinical practice, can help clinicians become familiar with these algorithms and consider their use in providing personalized therapy.

## 2. Dissecting the TME with Multi-Omics Approach

It is well known that the tumor microenvironment (TME) is favorable for cancer onset and tumor evasion, leading to drug resistance, especially for immunotherapeutic agents. In particular, the tumor immune microenvironment (TIME), a part of the TME, plays a crucial role in influencing the treatment response and can be categorized as immune-inflamed, immune-excluded, and immune-desert on the basis of the TILs number and proximity to tumor cells, determining the immunotherapy response [1,6,7]. As a result, extensive research is constantly being conducted to identify new targets for tailored therapies and biomarkers to predict treatment responses. In this regard, multiple algorithms have been developed over the years, exploiting the multi-omics perspective to achieve these goals (Table 1).

ESTIMATE, xCell, MCP-counter, and CIBERSORT are algorithms performing gene set enrichment analysis (GSEA) in RNA-seq data. Immunomics technologies can identify potential biomarkers of immune checkpoint blockers (ICBs), and transcriptomic data can be used to estimate the immune cell composition in the TIME [1].

Zeng and his colleagues developed a computational tool to support Immuno-Oncology Biological Research (IOBR). This tool comprises CIBERSORT, ESTIMATE, quanTIseq, TIMER, IPS, MCPCounter, xCell, and EPIC. IOBR optimizes precision immunotherapy by analyzing the TME, genomic alteration landscapes, and the latent correlation between patient stratification and therapeutic sensitivity and intervention [8,9,10,11,12,13,14,15,16].

Similarly, Jiang et al. developed the Tumor Immune Dysfunction and Rejection or Exclusion Score (TIDE), an analytical tool based on an association of gene signatures estimating cancer immune evasion that can predict the efficacy of immune checkpoint blockades [17].

In a recent study, Qin and his group used the TIDE, showing that enhanced TIDE scores correlate to a higher expression of migrasomes—a novel, identified, large vesicle containing chemokines, cytokines, and growth factors able to promote tumor metastasis and immune escape, reducing the response to ICI. Additionally, the researchers also evaluated ESTIMATE scores of migrasomes in different tumors, finding out that patients treated with immunotherapy had higher scores, which correlates with a better prognosis [18].

On the other hand, Zhu et al. have identified three subtypes with different TIDE scores and unique tumor microenvironments (TMEs), based on the expression of 66 prognostic genes. This subtype classification can predict the response to immunotherapy, with the subtype that has a lower TIDE score showing a better response to PD1/CTLA4. In addition, researchers have discovered an association between the mRNA-based expression subtypes and ICI response, while the Tumor Mutational Burden (TMB) status and transcriptome expressions can be used to guide treatment choices using platform L1000. Also, they created a classifier that can help identify the best drug for a colon cancer patient based on their gene expression [19].

Focusing on treatment sensitivity, Yuan et al. led a study using the R package pRRophetic (R version 4.3.3). The study predicted immunotherapy responses by analyzing gene expression profiles using the ImmuneCellAI algorithm (https://github.com/lydiaMyr/ImmuCellAI, accessed on 18 March 2024). By using TIDE and The Cancer Immunome Atlas (TCIA) platforms (https://tcia.at/home, accessed on 18 March 2024), the Immunophenoscore (IPS) (https://tcia.at/tools/toolsMain, accessed on 18 March 2024) and exclusion scores were provided. Better responsiveness to immunotherapeutic agents is achieved when the IPS is higher, and the exclusion score is lower [20].

Shi et al. proposed a Gluco-Immune Score by analyzing the TME where gastric cancer (GC) patients with high scores are beneficiaries of IT [21] as well as the use of a gene-based antigen processing and presentation signature (APscore) suggested by Wang et al. [22]. The importance of TME in affecting responses to ICI were also underlined by Zeng at al. who validated the TMEscore in a prospective phase II study that enrolled metastatic GC patients receiving pembrolizumab [23].

On the basis that immunotherapy efficacy depends on immune cell amount, the spatial relationships, and the TME conformation, Tong et al. retrospectively studied intertumoral CD8+ TILs in PD-L1-negative GC patients as a predictive factor for chemoimmunotherapy responses [24,25,26].

Also, tertiary lymphoid structures (TLS), part of the TME, correlate to better immunotherapy outcomes in GC patients [27].

Pooling together TMEscores and NanoString technologies, Zeng et al. [23] investigated the multi-omics data of 1524 patients with gastric cancer, then conducted a prospective, open-label, phase II trial (NCT02589496) employing mGC patients treated with pembrolizumab, suggesting that patients classified as Epstein Barr Virus (EBV) positive and Microsatellite Instability High (MSI-H) are the most responsive with higher TMEscores. Based on such results, the authors validated the robust predictive role of the TMEscore in the response of ICIs alone or combined with chemotherapeutic or antiangiogenetic agents. To make the TMEscore a useful clinical immunotherapeutic biomarker, Zeng et al. conducted two more clinical trials on gastric cancer patients IT-treated (NCT04850716 and NCT04850729). Features employing the *ARID1A* and *PIK3CA* alterations, kynurenine, glycogen metabolism, *ATG7*, and *VAMP8* methylation shed light on putative mechanisms of TMEscore-guided precision immunotherapies [23].

Following the example of Zheng’s group, further study design involving multi-omics should be encouraged.

### Omics Sciences (OSc), Microbiome, and TME

It is well documented that the microbiome influences the immune system cells in the TME, leading to a pro- or anti-tumoral phenotype.

Anagnostou and colleagues observed a correlation between genomic alteration and loss of mutation-associated neoantigens in resistant tumors, leading to less treatment benefits. Similarly, focusing on non-small cell lung cancer, Duruisseaux and colleagues created the EPIMMUNE signature, which identified specific patterns of DNA methylation from nivolumab- or pembrolizumab-treated patients and was associated with clinical benefit.

As for potential microbiome-linked biomarkers for the response prediction of ICIs, indole, aldehydes, and short-chain fatty acids have emerged thanks to novel techniques and more accurate information about the interaction between the microbiome and ICIs [28,29,30].

A recent field of research leads to the role of the microbiome in affecting the lymphocytes activity. A study led by Zhang et al. [31] has examined the correlation between the microbiome found within pancreatic tumors and various clinical features such as prognosis, tumor microenvironment heterogeneity, and response to treatment. The study divided patients into two clusters, A and B. The former is associated with a poor outcome and worse staging and grading in comparison to cluster A. Immune analysis found a notable increase in immune infiltration in cluster B. In contrast, cluster A patients were more likely to benefit from CTLA-4 blockers [3].

Data available regarding multi-omics and the microbiome are few, suggesting that there is still work to be done.

## 3. Immunotherapy and Biomarkers: Classical Perspective

Since anti-PD1 and anti-PD-L1 have been approved and used in clinical settings, researchers focused on identifying biological molecules indicators of response.

The PD-L1 expression level was the first predictive biomarker. Although detectable via IHC with four FDA authorized diagnostic tests—22C3, 28-8, SP142, and SP263—the evaluation of PD-L1 expression employs IHC indicators such as the combined positive score (CPS) and tumor proportion score (TPS). TPS represents the percentage of tumor cells stained with any intensity of PD-L1, while CPS accounts for PD-L1-positive tumor cells and tumor-associated immune cells divided by the total number of tumor cells. Moreover, the analysis of the PD-L1 expression in circulating tumor RNA (ctRNA) conducted by Ishida et al. suggested the utility of ctRNA for predicting and monitoring immunotherapy responsiveness [32,33].

Other well-known biomarkers are the Tumor Mutation Burden (TMB) and Tumor Infiltrating Lymphocytes (TILs). The TMB generates a considerable number of neoantigens, which can trigger immune responses with consequent better prognosis, while the intensity of TILs strongly correlates with the response to immunotherapy and clinical outcomes. To note, the TMB’s accuracy has few limitations because not all mutations lead to new antigens formation and are subdue due to factors like tumor type, exposure to external carcinogens, detection methods, and genetic mutations in the TME [1,6,7].

Several studies have demonstrated that MMR status can be used as a predictor of response for patients undergoing immune checkpoint inhibitor therapy. Specifically, deficient MMR status, also known as microsatellite instability (MSI-High), is involved in the onset of various cancers and in a favorable response to immunotherapeutic agents [2]. Moreover, multiple studies have found a positive correlation between a high count of T-cell infiltration and an improved response to immune checkpoint blockade treatment [2].

Regarding MSI and TMB, recent studies have shown that high MSI and TMB relate to tumor antigenicity and ICI response. Evrard et al. found that screening for dMMR/MSI along with TMB may be helpful to determine the benefits of IT in CRC. Vanderwalde et al. established a link between TMB, PD-L1, and MSI through NGS. Previously, a tool was developed (available online) to predict IT responses by employing large-scale data from omics, CRISPR screening, studies concerning immunotherapeutic agents, and tumor profiles not related to IT [28,29,30].

Recently, Li and colleagues helmed a comprehensive analysis of multiple cancer types. The study found an abnormal expression of SERPINE1 (serine protease inhibitor) in cancer cells that correlates with immunoregulator expression, TMB, MSI, immune cell infiltration, and IT response. These findings highlight the importance of SERPINE1 and its potential implications in tumor immune escape, providing valuable insights that could guide individual, tailored, personalized treatment [3].

The actual usefulness of these predictors is limited by the scarcity of standardized and uniformed methods to detect PD-L1, the estimation of NGS with dissimilar sequencing panels, and TMB heterogeneity depending on the laboratory and platform used [34].

### Immunotherapy and Biomarkers: The New Perspective

As technology advances, new perspectives have opened up. With the increasing use of biological and computational technologies, multi-omics cancer data are now available, such as TCGA (The Cancer Genome Atlas), which provides data on somatic mutations; copy number variations (CNVs); DNA methylation profiles; and gene, microRNA (miRNA), long non-coding RNA (lncRNA), and protein expressions for 33 cancer types [35].

Lin et al. [34] have developed a tool called CAMOIP, standing for Comprehensive Analysis on Multi-Omics of Immunotherapy in Pan-cancer, that helps to screen various prognostic markers essential to distinguish immune responders from non-responders, identify potential beneficiaries, and monitor adverse immune-related events. It evaluates the implication of a gene mutation or alteration and verifies the correlation mechanism between immunotherapy biomarkers and the affiliated treatment response, assessing if a specific gene expression level correlates with patient outcomes after receiving ICIs [3].

Multi-omics algorithms refer to pan-cancer data. However, due to the intra- and intertumoral heterogeneity, an in-depth cancer specific analysis is required.

In this regard, He et al. tried to identify biomarkers predicting the response to immunotherapy, analyzing the transcriptome and somatic alterations data files available on TCGA about melanoma (UCSC-Xena) (https://xenabrowser.net/datapages/, accessed on 11 March 2024). The study indicates that tumor immunity, intratumor heterogeneity (ITH), TMB, copy number alterations (CNAs), PD-L1 expression, immune signatures and pathways stimulating the immune system response, higher enrichment scores of differentiations, EMT, invasion, and metastasis signatures are linked to ICI response. The researchers also identified other biomarkers, such as the negative correlation between activated mast and dendritic cells signatures with IT response, as well as the positive associations between the enrichment of many oncogenic pathways (JAK-STAT, RAS, MAPK, HIF-1, PI3K-Akt, and VEGF pathways), the number of microRNAs and proteins expression, and responsiveness to immunotherapy, while the mTOR pathway negatively correlates with IT response [35].

Moreover, Hu et al. led a study in which they classified Triple Negative Breast Cancer (TNBC) patients based on their immune subgroups (IS 1, 2, 3A, and 3B). They found substantial diversity in prognosis, response to immunotherapy and chemotherapeutic agents, gene mutations, and the infiltration of immunity cells. This study provides the groundwork for the use of this categorization to foresee the IT potential in this type of cancer [36,37].

Furthermore, the TCIA database (https://tcia.at/home, accessed on 18 March 2024) provides immunotherapy prediction information based on comprehensive immunogenomic analyses of next-generation sequencing data (NGS) of 20 types of cancer, including breast cancer. This information can be used to predict the immune checkpoint inhibitor (ICI) response of patients with distinct subtypes of breast cancer [19].

In some cases, OSc was used to define the molecular subtypes of GC like in the Hu et al. study that identified CS1 and CS2 GCs with CS2 resulting as the one that benefited the most from ICI therapy [38]. Meanwhile, Yuan et al. found out that HER2-related metabolic heterogeneity in GC is linked to ICI response in neoadjuvant chemotherapy, showing that the quiescent, aspartate, and glutamate subtypes are better responders [20].

Moreover, a meta-analysis evaluating ten different tumor types highlighted the role of multiple immunofluorescence (mIHC/mIF) as a predictive factor of ICI response over TMB, gene expression profiling, and PD-L1 expression [39].

Also, Chen et al. confirmed the role of the TIL signature found through mIHC and multi-dimensional analyses in foreseeing anti-PD-1/PD-L1 responsiveness [33,40].

On the other side, based on low- and high-TMB, Fu et al. proposed an immune prognostic model based on a side-by-side of gene expression and immune markers [41].

Summing up, several researchers employed multi-omics approaches to evaluate IT activity and responsiveness and explore new targets for combinatorial treatment.

However, in the field of cancer immunotherapy, transcriptomics is generally used with proteomics to showcase phenotype differences, but only a few studies focus on ICI’s added metabolomics, limiting the capability to actively translate the use of these methods into clinical practice [42]. Therefore, an extensive multi-omics approach is desirable. 

## 4. In-Depth Cancer-Type Analysis: Focus on Clinical Trials

Basically, OSc has the goal of helping physicians to understand the complex mechanism behind response or resistance to ICI therapies [43,44,45,46,47,48,49,50,51,52,53,54,55,56]. There are a few studies analyzing specific tumor-leveraging, multi-omics methods (Table 2).

Song et al. [47] explored ITH, known for interfering with the response to ICI, in a phase II multicenter clinical trial that enrolled NSCLC patients treated with NK046 at the multi-omics spatial level [46], showing the role of a “non-responder” stromal area in affecting the responses to ICI, suggesting that spatial segmentation is the key to being more precise and predicting the effectiveness of biomarkers while considering the ITH [47].

But, apparently, among the cancers, GC is the most studied. Currently, the main recognized predictors of ICI response are the tumor proportion score (TPS) and combined positive score (CPS). Their role has been investigated in several clinical trials such as KEYNOTE-012, KEYNOTE-059, CheckMate032, and ATTRACTION-2. The TPS together with the blood-neutrophil-to-lymphocyte ratio (NLR) and serum Na levels can be used as biomarkers to forecast responsiveness to anti-PD-1 in pre-treated GC patients. The same results were obtained in the KEYNOTE-061 trial for Pembrolizumab [48].

Furthermore, an exploratory analysis using RNA seq observed a correlation between the T-cell-inflamed gene expression profile (TcellinfGEP) with the Objective Response Rate (ORR) and Progression-Free Survival (PFS), specifically for pembrolizumab. The UMIN000025947 study (phase I/II) investigated a combined treatment (nivolumab, paclitaxel, and ramucirumab) used after the first line along with a phase II trial held by the Arbeitsgemeinschaft Internistische Onkologie (AIO), highlighting the CPS’s prognostic value in predicting the ICI response [18]. Similarly, the MAHOGANY study explored the combination of margetuximab (anti-HER2) and pembrolizumab (anti-PD-1), confirming the role of CPS as a predictive score [48].

However, the CPS cut-off for IO response predictivity is still debated [49,50]. Nevertheless, various studies suggest a CPS score of 1 as the threshold for ICI monotherapy’s survival advantage; few analyses, including CPS scores of 10 in KEYNOTE-059, KEYNOTE-061, and KEYNOTE-062 trials concerning GC patients, have steadily demonstrated favorable outcomes obtained with pembrolizumab, despite the treatment lines in which it is used [50].

Historically, MSI-High status is considered the best predictor of ICI response. A trial analyzing GC patients treated with anti-PD-1 (nivolumab) suggested the use of CPS and MSI as predictors of treatment response [51]; however, according to an extensive meta-analysis that included studies like CheckMate-649, KEYNOTE-061 and -062, and JAVELIN Gastric 100, around 50% of MSI-H tumor patients with gastric cancer are intrinsically resistant to PD-1 inhibitors.

An explanation came from Kwon and colleagues who suggested a link between prolonged PFS and the T-cell receptor repertoire under treatment with pembrolizumab. The increase in the amount of double-positive T cells (CD8+/PD-1+) in MSI-H GC patients is associated with lasting clinical profit [52].

Regarding the combination of old and new techniques and biomarkers, in a phase II exploratory study held by Song et al., the neoadjuvant combination of camrelizumab (anti-PD-1), apatinib (antiangiogenic), and chemotherapy was evaluated. The trial used sequential multi-omics techniques to highlight potential biomarkers for neoadjuvant immunotherapy; they observed correlations between pathological responses and MSI status, CPS, and TMB. Another phase II trial called Neo-PLANET investigated neoadjuvant camrelizumab with concurrent chemoradiotherapy for locally advanced GC/GEJC. The trial found a positive correlation with the TMB median level (4.04 mutations/Mb) estimated before the treatment onset [53,54].

The NCT02915432 trial (phase I/II) evaluated an anti-PD-1, toripalimab, and showed that patients with TMB-high had better outcomes. Also, a preliminary investigation of KEYNOTE-061 unveiled a solid relation between TMB and pembrolizumab effectiveness when used in the second line, as in the EPOC1706 trial investigating pembrolizumab, in association with lenvatinib in the first- or second-line setting.

On the other hand, MSI-H/dMMR gastric cancer patients with low TMB are less responsive to anti-PD-1 agents, as shown in the study of Chida et al.; so, Wang et al. proposed to screen responders to ICI based on the combination of blood MSI and blood TMB [55,56].

Further studies employing a multi-omics strategy focusing on distinct cancer types are also desirable.

## 5. ICI-Induced Immune-Related Adverse Events (irAEs): The Other Side of the Coin

Despite the well-known benefits of ICI treatment, there are also autoimmune side effects to consider such as irAEs. Their frequency varies depending on the ICI used, but it may account for up to 70% of cases. These side effects can involve any organ and can appear at any point during treatment, even several months after stopping ICI. Severe irAEs (grades 3–4) may require hospitalization, pausing or stopping treatment, and immunosuppressant medication. In some cases, irAEs can be fatal [57,58].

Grigoriou et al. conducted a study on the changes in the T-regulator cells in blood and found that the reprogramming of the inflammatory T-regs could be a potential indicator for the development of irAEs. Das et al. helmed a study, using scRNA seq, showing that early modifications in B-cells could mark the irAEs risk in melanoma [42,59,60,61].

Jing and colleagues used a model to identify two potential predictive biomarkers—lymphocyte cytosolic protein 1 (LCP1) and adenosine diphosphate dependent glucokinase (ADPGK)—involved in T cell activation. The study also found that six factors correlate to irAEs as well as a better IT response (TMB, T-cell receptor (TCR) diversity, interferon (IFN) α level, tumor necrosis factor (TNF) α, eosinophils, and neutrophils). Among these factors, CD8+ T cells and TCR diversity showed the highest predictive effectiveness [58,62].

Nuñez et al. led a study on patients with melanoma or NSCLC who were receiving ICIs treatment, using a multi-omics approach to look for immune signatures. By combining the obtained data, the researchers were able to identify potential predictive biomarkers of ICI-related irAEs, observing an early increase in CXCL9/CXCL10/CXCL11, IFN-g, and Ki-67+ T cells that may indicate an increased risk of the occurrence of irAEs. Also, an increase ahead of time of proliferating T-cell subgroups occurred mainly in patients who experienced ICI-related toxicity. Patients who had higher frequencies of CD8+ CD38+ Ki-67+ T cells at certain time points were more likely to develop autoimmune toxicity sooner. Therefore, combining cellular and proteomic biomarkers can support clinicians to identify which patients will benefit the most from ICI therapy and those who are most likely to develop irAEs, requiring increased surveillance [57].

## 6. AI: The New Frontier

Apart from multi-omics, AI is currently a hot topic in each research field. To leverage the power of AI to predict patient responsiveness to immunotherapeutic agents, it is necessary to set up a training and validation cohort, collecting medical data, which include pathological tissue, CT/MR imaging-omics, genomics, proteomics, and more, from the training cohort by filtering, segmenting, extracting, and selecting features and then handing them over to the AI for learning and modeling. Finally, the validation cohort is used to verify the learning results. AI can also analyze pathological data and determine the state of tumor development. Its algorithm and analysis can be easily standardized and shared.

Several researchers have made this effort over the years.

In fact, Bojar et al. used a serum proteomics model implemented by AI to foresee the IT response [63]. Meanwhile, Xie and colleagues developed a predictive model to distinguish between “cold” and “hot” immune patients. This model was applied to analyze clinical data, showing a better IC response in patients with high immunogenicity [64]. Similarly, Gupta and colleagues used a probabilistic model called the Bayesian network, combining anamnestic data and tumor features to predict survival outcomes in patients with RCC undergoing nivolumab [65].

On the other hand, Hu et al. demonstrated that a tumor proportion score (TPS) analyzer based on AI can increase reliability, leading to an accurate prediction of the IT response in NSCLC [66].

Yi Yang and colleagues developed a deep learning model including anamnestic, clinical, laboratory, and imaging data of NSCLC patients [67]. Likewise, researchers using a deep learning system discovered predictive signatures and radiomics markers for progressive disease, survival, and IT responsiveness forecasting in patients with NSCLC or melanoma [68,69,70,71,72,73,74,75,76,77,78,79,80,81].

Prelaj et al. analyzing real-world data to predict outcomes in patients that respond or do not to treatment using deep learning techniques. They even reviewed several studies using AI tools to confirm classical and new biomarkers like human leukocyte antigen loss of heterozygosity (HLA LOH) and genomic ITH for IO response [82].

Also, Harder and his group detected the expression level of PD-L1 in a non-invasive way, combining radiomics, based on CT, with clinical characteristics [83].

Furthermore, Peng Song and his group theorized that combining clinical and laboratory data with the ones obtained from DNA and RNA seq, as well as immunostaining, would help to assess therapy effectiveness [84].

Sun and colleagues led a study establishing a predictive imaging model of patients’ outcomes after immunotherapy, combining CT scans with RNA-seq data obtained from cancer biopsies [85].

Also, new RNA-based biosignatures, generated using AI transcriptomics and DNA methylation profiles, have been found to have potential in predicting the ICI response across cancers [86,87,88,89,90,91].

Ultimately, the NSCLC study found that using a model including different sources of data resulted in better performance than models that used single data. Also, Yang et al. showed that the use of a model including radiomics and clinical and laboratory data of metastatic NSCLC patients was more accurate compared to traditional evaluations [92,93,94].

### The AI Tools: Pros and Cons

Discovery tools powered by AI have assisted in identifying new immune phenotypes related to immunotherapeutic agents’ responsiveness.

Medical datasets used for AI-based research can be challenging to work with due to their complexity, high dimensionality, noisiness, and incompleteness. To achieve high quality, a vast volume of excellent data is required. Currently, there are numerous ML-based models available, making it difficult to choose the appropriate methodology. Standardization, protocols, and guidelines like CONSORT-AI, SPIRIT-AI, and the ML-CLAIM checklist provide AI-specific recommendations, but many studies lack clear documentation [95,96,97].

Additionally, the complexity of the algorithm can be solved using XAI methods. For generalizability and robustness, ML models require exposure to diverse data sources. The studies reviewed lacked endpoints and pipelines. Therefore, future studies should be well designed as prospective studies, choosing the most appropriate AI approach—an example is represented by the I3LUNG study (NCT05537922)—or as observational studies guided by data, especially for biomarker-driven detection, like in the NCT0555096 study and the so-called APOLLO 11 trial [82].

## 7. Conclusions

The emergence of cutting-edge technologies such as scRNAseq, high-parameter flow cytometry, and spatial transcriptomics has revolutionized our understanding of anti-tumor immune responses. As these technologies become more advanced and widely adopted in the future, they have the potential to provide us with profound insights into the behavior of immune cells within the tumor microenvironment (TME). This knowledge could serve as the basis for designing more effective immunotherapies. However, to achieve this goal, it is essential to pair the vast amount of data produced by these technologies with focused experimental questions, precise analysis, clear interpretation, and supportive mechanistic studies. To accurately identify new targets for immune checkpoint inhibitors (ICIs), it is crucial to determine the antigen specificity within the TME [57].

Multi-omics data integration approaches dissecting every functional layer of different cell types unravel the underlying pathophysiological mechanisms of cancers, facilitating tumor classification, diagnosis, and prognosis [20].

That said, if scientists focus not only on the classical pathological features of the tumor but also on the underlying tumor biology along with the inter- and intratumoral heterogeneity, while also being open to welcoming and imbricating new technologies with the old ones (i.e., sharing data and codes), it will make a difference in designing studies leading to practice-changing and to better outcomes.

## Figures and Tables

**Table 1 ijms-25-03563-t001:** Algorithms and where to find them.

Algorithm Name	Website	Object of Investigation
ESTIMATE	https://bioinformatics.mdanderson.org/public-software/estimate/, accessed on 11 March 2024	Stromal Score (detecting the stroma), Immune Score (the infiltration of immune cells) and Estimate Score (that infers tumor purity) in tumor tissue
xCell	https://github.com/dviraran/xCell, accessed on 11 March 2024	Cell type enrichment analysis from gene expression data for 64 immune and stroma cell types
MCP-counter	https://github.com/ebecht/MCPcounter, accessed on 11 March 2024	Estimating the population abundance of tissue-infiltrating immune and stromal cell populations using gene expression
CIBERSORT	https://cibersortx.stanford.edu/, accessed on 11 March 2024	Estimation of the abundances of member cell types in a mixed cell population, using gene expression data
IOBR	https://iobr.github.io/IOBR/IOBR-VIGNETTE.html, accessed on 11 March 2024	Decode tumor microenvironment and signatures
TIDE	https://github.com/jingxinfu/TIDEpy?tab=readme-ov-file, accessed on 11 March 2024	Potential of immune escape
TMEscore *	https://github.com/DongqiangZeng0808/TMEscore, accessed on 11 March 2024	TME infiltration patterns were determined and systematically correlated with TME cell phenotypes, genomic traits, and patient clinicopathological features

* The only score used in clinical trials.

**Table 2 ijms-25-03563-t002:** Overview of the Clinical trials investigating IT biomarkers.

Clinical Trial	Phase	Tumor Type	Biomarkers	Omics Methods
Song et al. [47] (NCT03838848)	II	NSCLC	Stroma	Multi-omics spatial level
Song et al. [47] (NCT03878472)	II	GC	MSI CPS TMB	Sequential multi-omics
KEYNOTE-012 KEYNOTE-059 KEYNOTE-062 CheckMate032 ATTRACTION-2	Ib II III I/II III	GC	TPS CPS	-
KEYNOTE-061	III	GC	TPS NLR Na level	-
UMIN000025947 AIO trial MAHOGANY KEYNOTE-061	I/II II II/III III	GC	CPS	-
CheckMate-649 KEYNOTE-061 KEYNOTE-062 JAVELIN Gastric 100	III III III III	GC	CPS MSI	-
Neoplanet NCT02915432 KEYNOTE-061 EPOC1706	II I/II III II	GC	TMB	-
Chida et al. [55] Wang et al. [56]	Exploratory study -	GI GC	MSI TMB	- NGS-based

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
