# Peer review of "Immunotherapy and Cancer: The Multi-Omics Perspective"

_ijms, 2024, doi:10.3390/ijms25063563_

Round 1

Reviewer 1 Report

Comments and Suggestions for Authors

Line 166-167: Authors state that there is negative correlation between immunotherapy responses and dendritic cells. Which type of dendritic cells? In general, this is contrary to the literature, which supports that Batf3 DC otherwise known as cDC1 have positively correlated with immunotherapy responses (such as checkpoint inhibitors and other agents such as STING agonists. Could you please clarify?

Author Response

Thank you very much for your report. Line 210-212 (former 166-7): As for the DC type, the authors didn't do the subcluster analysis. They observed and reported a negative correlation between the activated signature of the DCs and the IT response. On your suggestion, I rephrased the sentence to be clearer.

Reviewer 2 Report

Comments and Suggestions for Authors

The review titled "Immunotherapy and Cancer: A Multi-Omics Perspective" authored by Donisi et al. aims to investigate novel models for identifying biomarkers that contribute to patient stratification and personalized treatment, emphasizing their potential pivotal role in transforming clinical practice.

A good number of review articles on this subject have been published in the last three years, as evidenced by PubMed, Reaxys, and SciFinder search. Can the author briefly highlight the relevance and use of this review. What was their criteria on selecting the articles they included in their review.

To improve the presentation of the content and enhance clarity, the authors are suggested to incorporate figures and tables. Notably, Figure 1 is referenced in Section 3, line 96, but is missing in the current manuscript.

The manuscript's flow could be improved. While the initial focus appears to be on the tumor microenvironment or tumor immune microenvironment, the narrative becomes somewhat scattered, touching upon biomarkers and clinical studies without a clear structure.

A suggested revision involves reorganizing the sections to center around the microenvironment and subsequently delve into biomarkers, creating a more cohesive narrative. Then a separate section on clinical relevance or studies. A table would be highly useful in that section.

Additionally, it is recommended to introduce a section on post-translational modifications, as these processes elicit immune responses that could be valuable for identifying cancer biomarkers. This addition would contribute to a more comprehensive understanding of patients' outcomes and strengthen the overall structure of the review.

Author Response

Thank you very much for your constructive report.
We tried, as suggested, to briefly highlight the relevance and use of the review and which types of articles we included in the introduction. Moreover, we add a blurb about post-translational modifications in the introductory paragraph.
There's no Figure 1 referenced anymore. We incorporate two tables to improve clarity.
We also attempt to improve the manuscript's flow, reorganizing the main structure and sections.